# Porous V_2_O_5_/TiO_2_ Nanoheterostructure Films with Enhanced Visible-Light Photocatalytic Performance Prepared by the Sparking Method

**DOI:** 10.3390/molecules25153327

**Published:** 2020-07-22

**Authors:** Porntipa Pooseekheaw, Winai Thongpan, Arisara Panthawan, Ekkapong Kantarak, Wattikon Sroila, Pisith Singjai

**Affiliations:** 1Department of Physics and Materials Science, Faculty of Science, Chiang Mai University, Chiang Mai 50200, Thailand; orm.kalo@gmail.com (P.P.); winai.thongpan@gmail.com (W.T.); vampire_601@hotmail.com (A.P.); ekkapong_k@hotmail.com (E.K.); ple_wasan@hotmail.com (W.S.); 2Ph.D’s Degree Program in Applied Physics, Faculty of Science, Chiang Mai University, Chiang Mai 50200, Thailand; 3Center of Excellence in Materials Science and Technology, Chiang Mai University, Chiang Mai 50200, Thailand

**Keywords:** V_2_O_5_/TiO_2_ nanoheterostructures, porous films, sparking method, photocatalytic activity

## Abstract

Porous V_2_O_5_/TiO_2_ nanoheterostructure films with different atomic ratios of Ti/V (4:1, 2:1, 1:1, and 1:2) were synthesized by a sparking method for the first time. The sparking method, which is a simple and cost-effective process, can synthesize highly porous and composite films in one step. Field-emission scanning electron microscope (FE-SEM) images revealed the porosity morphology of all prepared samples. V_2_O_5_/TiO_2_ nanoheterostructure films were confirmed by Raman spectroscopy, high-resolution transmission electron microscopy (HRTEM), and X-ray photoelectron spectroscopy (XPS). The secondary particle size and band gap of the samples were highly correlated to the V_2_O_5_ proportion, resulting in enhanced visible-light absorbance. V_2_O_5_/TiO_2_ nanoheterostructure films at an atomic ratio of 1:1 showed the highest photocatalytic performance, which improved the degradation rate up to 24% compared to pure TiO_2_ film. It is believed that the formed nanoheterostructure and greater portion of V^4+^ ions are reflected by this ratio.

## 1. Introduction

Titanium dioxide (TiO_2_) is a widely used photocatalyst because of its low cost, environmental friendliness, and stable photocatalytic reaction [1,2]. There are three forms of TiO_2_ phases (anatase, rutile, and brookite), in which the anatase phase has higher photocatalytic activity than both the rutile and brookite phases. Moreover, a mixed phase of anatase and rutile was found and is a better alternative [3,4]. However, the wide energy gap of TiO_2_ (~3.2 eV) limits use of the visible-light region as the excitation energy for photo-generated electron-hole pairs. At present, there are many methods that have been used to improve the photocatalytic property of TiO_2_.

Surface modification is known to be able to promote photocatalytic performance by providing more effective interfacial properties. In particular, the large surface areas of 1-D structures and porous morphology are preferred in the design [5,6]. Moreover, composite films with a narrow band gap semiconductor are preferred, which can lead to superior performance of the pure TiO_2_ films [7,8,9]. Both TiO2 and V_2_O_5_ are two important materials due to their excellent electronic, chemical, and optical properties [10,11]. The strong interaction between TiO_2_ and V_2_O_5_ is a common scientific phenomenon used in catalytic systems. Furthermore, the heterojunction between V_2_O_5_ and TiO_2_ can improve the photocatalytic efficiency of the host by supporting the separation and transfusion of photogenerated electron-hole pairs at the interfaces [12,13]. In addition, transformation of oxygen vacancies during preparation processing can further promote the separation of photoexcited electrons and holes [14,15].

However, the simplicity and cost-effectiveness of the production method are important factors to consider when applied to large-scale manufacturing. At present, many physical and chemical techniques have been used for preparing V_2_O_5_/TiO_2_ composite films, including sol-gel, chemical vapor deposition, DC arc plasma, electrospinning, hydrothermal, and pulsed laser deposition [16,17,18,19,20,21]. Some techniques require multiple processes for fabricating porous structures and composite materials, which usually leads to expensive and complicated processes. The sparking method has the potential to synthesize highly porous and composite films in a one-step process. Using this process, films can be deposited directly onto a substrate without the need for a vacuum system [22,23]. Moreover, it is also possible to fabricate the films in large-scale manufacturing.

Therefore, photocatalytic performance can be effectively improved through the designs of porous morphologies and composite films. Moreover, simplicity and cost-effectiveness are also required for the production method. Thus, this paper aimed to synthesize porous vanadium pentoxide/titanium dioxide (V_2_O_5_/TiO_2_) nanoheterostructure films using the sparking method. The influences of the structural, morphological, optical, and photocatalytic properties of the prepared films were investigated and reported in this work.

## 2. Results and Discussion

### 2.1. Morphology, Crystal Structure, and Property Investigations

FE-SEM images revealed the porosity morphology of small particles for all prepared samples, which is the highlight of the sparking method. Our previous papers reported that the as-deposited particles produced by the sparking method were small with primary sizes of approximately 1–5 nm [24]. In this paper, the TiO_2_ films exhibited interconnected, spherical-like shapes with average secondary particle sizes of 20 nm that were measured by an imageJ program. While the composite films showed the particles, the sizes increased slightly when the proportion of vanadium increased due to the large size of V_2_O_5_ particles, as summarized in Table 1. The film′s thickness was approximately 300 nm for all prepared samples as confirmed by the cross-section analysis. In addition, the element composition of samples was determined using EDS and revealed that the atomic ratios of Ti/V were approximately 4:1, 2:1, 1:1, and 1:2 for TiV1-4 samples, respectively, as shown in the inset images of Figure 1b–e. Interestingly, the equal proportion of Ti and V wires in the TiV-4 sample did not correlate with the quantity of their atoms due to their own properties (melting point, latent heat, specific heat capacity) [25]. The sparked films in this experiment were very thin and porous; therefore, the large number of oxygen peaks in the EDS results is due to the quartz substrates (SiO_2_).

Figure 2 illustrates the Raman spectra of prepared samples in the range of 100 to 1100 cm^−1^. The peaks spiked at 144 and 631 cm^−1^, which correspond to E_g_ and B2_g_ vibration modes, respectively, and confirmed the anatase phase structure of the TiO_2_ film [26]. In the case of the V_2_O_5_ film, nine strong peaks of V_2_O_5_ (145.1, 283.2, 306.5, 406.7, 484.3, 521.9, 696.9, and 990.1 cm^−1^) and two weak peaks of VO_2_ (195.6 and 611.2 cm^-1^) were observed indicating the V_2_O_5_-VO_2_ mixed phase [27]. The inset image in Figure 2 shows a high-magnification of the overlap between the strong TiO_2_ (anatase) and main V_2_O_5_ (orthorhombic) peaks, which shifted from 144 to 145 cm^−1^ for TiV1-4 samples, respectively. The slight shift of peak positions could plausibly be regarded as the result of modifications in the force constants of tetragonal TiO_2_ lattices by the added V_2_O_5_, which indicates the formation of a V_2_O_5_/TiO_2_ nanoheterostructure [28].

The XPS survey spectra presented various peaks corresponding to Ti, V, O, Si, and C, which are elements of the prepared films, except only C occurs during the sparking method and Si from quartz substrate, as shown in Figure 3a. There were no impure peaks, which confirms the formation of pure V_2_O_5_/TiO_2_ composites films. The different core levels of high-resolution XPS spectra for Ti 2p, V 2p, and O 1s were investigated further. In Figure 3b, the two peaks centered at binding energies of 459.3 and 464.9 eV were assigned to be Ti 2p_3/2_ and 2p_1/2_ of Ti^4+^, respectively, which are consistent with the typical values of TiO_2_ [29]. In addition, the small shift of the Ti 2p peak position towards a higher binding energy indicates the integration of V ions into the TiO_2_ lattice [28]. The V 2p spectra displayed that V^5+^ and V^4+^ ions were uniformly distributed on the composite films, which were centered at 517.8 and 516.7 eV, respectively, as shown in Figure 3c. This affirms the result from Raman spectra that there was some content of VO_2_ in the composite films. In Figure 3d, three peaks at binding energies of 530.2, 531.1, and 533 eV observed in O 1s spectra also confirmed the O-Ti, O-H, and O-V bonding, respectively, of Ti-O-V bonds in the composite samples [30,31].

Normally, V^4+^ ions can only integrate within the TiO_2_ lattice by the replacement of Ti^4+^ ions because of the similar radii of V^4+^ and Ti^4+^ species. The Ti-O-V bond was formed by sharing oxygen atoms of the V^4+^ ions in the TiO_2_ lattice, leading to oxygen vacancy in the lattice and exposure to a high electron generation capacity. Moreover, we know that the greater number of V^4+^ ions led to the formation of more O_2_ radicals, which is the strongest oxidizing species in photocatalysis [32,33]. Thus, the existence of V^4+^ plays an important role in improving photocatalytic performance [34], which shows its maximum content in the TiV-3 sample as present by the V^4+^/ V_total_ ratio in Figure 4b. The V^4+^/V_total_ ratio were calculated by area under the curve of high-resolution V 2p XPS spectra as shown in Figure 4a.

The TEM and HRTEM images in Figure 5a,c confirm that the primary particle size of V_2_O_5_/TiO_2_ nanoheterostructure films was less than 20 nm and then agglomerated to secondary particles at larger sizes correlating to the SEM result. These images clearly present V_2_O_5_ particle interfaces that are dispersed on the surface of TiO_2_ particles, resulting in the formation of a heterostructure. The lattice fringes of 0.35 and 0.43 nm correspond to the (101) lattice distance of anatase TiO_2_ and the (001) orthorhombic V_2_O_5_, respectively, [30,31] and were measured to confirm the heterostructure at their particle interface, as shown in Figure 5d. These heterostructures are expected to adjust the charge separation and improve the photocatalytic efficiency. Moreover, Figure 5b represents the indexed selected area electron diffraction (SAED) patterns of the TiV-3 sample, which, again, confirms the presence of anatase-TiO_2_ and V_2_O_5_ phases and the absence of any other impurity [16].

### 2.2. UV–Vis Results, Band Gap Investigation, and Photocatalytic Performance

Figure 6a illustrates the UV-Vis absorption spectra of the prepared samples. Notably, large absorption in the visible-light region (peaks arise at 420 nm) was only seen in the V_2_O_5_, TiV-4, and TiV-3 samples, while other samples exhibited only UV absorption. Moreover, the band gaps (E_gap_) of samples were calculated from the Tauc plot as shown in Figure 6b [35,36]. The graph shows that the band gaps were approximately 3.28, 3.12, 2.84, 2.63, 2.56, and 2.32 eV for pure TiO_2_, TiV1-4, and pure V_2_O_5_ samples, respectively. These results show that the band gaps of the samples were slightly narrowed with increased vanadium proportion, and this led to enhanced photocatalytic activity under the visible-light region correlated with the previous absorption spectra.

The degradation rate of MB was calculated to investigate the photocatalytic activity under visible-light irradiation, as shown in Figure 6c. Calculations show that all composite films had more prominent photocatalytic performances than pure TiO_2_ that could be due to their narrowed band gap and the heterostructure at the interface between pure TiO_2_ and pure V_2_O_5_ particles [13,18,30]. To quantify the photocatalytic ability of the samples, the photocatalytic reduction process followed the pseudo-first-order kinetic reaction, which can be expressed as (ln(C_0_/C_t_) = kt), where C_0_, C_t_, k, and t represent the initial concentration of MB, the concentration of MB after visible light irradiation, the rate constant, and the reaction time, respectively [37]. As shown in Figure 6d, the k values were 0.0651, 0.0693, 0.0815, 0.0878, 0.0805, and 0.0544 h^−1^ for pure TiO_2_, TiV1-4, and pure V_2_O_5_ samples, respectively. Especially in the TiV-3 sample, the highest photocatalytic activity at 64.8% and maximum rate constant were achieved due to the large heterojunction interface and the existence of V^4+^ ions integrated within the TiO_2_ lattice [32].

### 2.3. Possible Photocatalysis Process in the Visible-Light-Irradiated V_2_O_5_/TiO_2_ System

To investigate the electronic band structure, the conduction and valence band edge position of TiO_2_ and V_2_O_5_ were calculated through the following formula [38]:(1)$$\mathrm{Conduction}\;\mathrm{band}\;\mathrm{edge}\;(E_{\mathrm{CB}})=X-E^{e}-\frac{1}{2}E_{g}$$
(2)$$\mathrm{Valence}\;\mathrm{band}\;\mathrm{edge}\;(E_{\mathrm{VB}})=E_{\mathrm{CB}}+E_{g\;}$$
where X is the absolute electronegativity of the semiconductor (TiO_2_ ~ 5.81 eV, V_2_O_5_ ~ 6.10 eV), E^e^ is the energy of free electrons on the hydrogen scale, and E_g_ is the band gap of the semiconductor [38]. When TiO_2_ and V_2_O_5_ semiconductors with different energy levels contact each other, electric charge is redistributed, leading to Fermi-level alignment. This induces electron and hole transfers at the V_2_O_5_/TiO_2_ interface. After combining TiO_2_ using V_2_O_5_, the band gap width was reduced and led to the absorption of more visible light, which is more conducive to dye degradation.

Figure 7 illustrates a schematic diagram of the possible photocatalysis process on the heterostructure of V_2_O_5_/TiO_2_ under UV-Visible light irradiation. The surface complex can be exited to generate photoelectrons (e^−^) and holes (h^+^) by V_2_O_5_ and TiO_2_ semiconductors. The e^−^ in the conduction band of V_2_O_5_, which has a higher valance band edge than that of TiO_2_, is produced by the difference in energy levels and shifts the conduction band of TiO_2_ in order to promote photoreduction and produce oxygen ion radicals (O_2_·^−^). The h^+^ at the valence band of TiO_2_ was produced and then transferred to the valence band of V_2_O_5_ to promote photooxidation. Furthermore, the presence of V^4+^, as evidenced by the XPS result in our work, supports and provides the higher active sites. It also provides high electrical conductivity, leading to boosting the separation and transfer of photogenerated charge carriers, which is important for photocatalytic reactions, as reported previously [39,40,41,42]. In this process, it was indicated that photoelectrons and holes were well separated and their recombinations were reduced. It is believed that the enhanced photocatalytic activity could be attributed to the synergism between efficient V_2_O_5_/TiO_2_ nanoheterostructures and the existence of V^4+^ ions.

## 3. Experimental Setup

### 3.1. Preparation of Samples

The experiment was carried out by sparking off four pairs of titanium wires (Ti, ø 0.25 mm, purity 99.5%, Advent Research Material Ltd., Oxford, UK) with a high DC voltage of ~3 kV and a limited current of ~3 mA for synthesizing pure TiO_2_ films. The spark pulse duration and repetition rate were fixed at 0.16 s and 6.2 Hz, respectively. The wires were then placed with 1 mm spacing and 1 mm above the substrate. The sparking head was adjusted to cover all substrates using computer numerical control (CNC) as shown in Figure 8a.

To synthesize V_2_O_5_/TiO_2_ nanocomposite films, the vanadium wires (V, ø 0.25 mm, purity 99.8%, Advent Research Material Ltd., Oxford, UK) and titanium wires were sparked at the same time. The vanadium concentration was varied by replacing titanium wires with vanadium wires from one pair to four pairs, which were labeled as TiV1-4, respectively, as shown in Figure 8b. Finally, vanadium and titanium nanoparticles were deposited onto the substrate and formed nanocomposite films with different atomic ratios of Ti/V. All as-deposited films were synthesized on quartz glass substrates (10 × 10 × 1 mm^3^) for 30 min at room temperature and post-annealed at 400 °C for 1 h at atmospheric pressure. The structural, morphological, optical, and photocatalytic properties of the samples were investigated and reported in this paper.

### 3.2. Characterization Methods

The surface morphology of samples was observed by a field-emission scanning electron microscope (FE-SEM, JEOL JSM 6335F, JOEL Ltd, Concord, MA, USA) equipped with an energy-dispersive spectroscope (EDS, Oxford, Concord, MA, USA) to study the film’s elemental distributions. The high-resolution transmission electron microscope (HRTEM) images were analyzed by a JEOL model JEM 2010 instrument (Concord, MA, USA) at an accelerating 200 kV. The structures of samples were investigated by a Raman spectrometer (JOBIN YVON HORIBA T64000, palaiseau, France) at room temperature with 514.5 nm of 150 mW Ar laser and X-ray Photoelectron Spectroscope (XPS, Rigaku Smartlab Japan system, Rigaku Corp., Tokyo, Japan) using an Al K-alpha source at 1487 eV. Optical properties were carried out over a wavelength range from 200 to 800 nm using a UV-Vis spectrometer (Perkin Elmer Instruments, Tokyo, Japan).

### 3.3. Photocatalytic Activity Measurement

The photocatalytic reaction of samples was evaluated by 0.01 mmol/L methylene blue (MB) solution (Ajex Finechem Pty. Ltd., Sydney, Australia) under visible light for 1–5 h. visible irradiation was produced by a 75 W Xenon lamp (100 w/m^2^, Philip, Bangkok, Thailand), which has a spectrum nearly similar to the solar light spectrum. The catalyst films were placed on the bottom of cuvettes with aqueous MB solution, and then variations in the concentration of MB with samples were analyzed at given irradiation times by a UV-Vis spectrometer (Perkin Elmer Instruments, Tokyo, Japan) in absorption mode. The degradation rate of MB, which represents the photocatalytic efficiency of films, can be calculated with the following Equation:(3)$$D\;\left(\%\right)=\left[\frac{C_{0}-C_{t}}{C_{0}}\right]\times 100$$
where D is degradation rate, C_0_ is absorption before irradiation, and C_t_ is absorption after irradiation [43].

## 4. Conclusions

Porous V_2_O_5_/TiO_2_ nanoheterostructure films with various atomic ratios of Ti/V (4:1, 2:1, 1:1, and 1:2 for TiV1-4 samples, respectively) were successfully fabricated by the sparking method in one step. Raman spectra, SAED patterns, and HRTEM were used to confirm the formation of V_2_O_5_/TiO_2_ nanoheterostructure films and their particle interface area. XPS revealed the existence of V^4+^ that integrated within the V_2_O_5_/TiO_2_ nanoheterostructure and had a significant effect on its photocatalytic performance. The interface heterojunctions on the surface of the catalyst are believed to encourage the photocatalytic performance through electron-hole pair production and the easy electron transport mechanism between the valance and conduction bands due to the low band gap of V_2_O_5_/TiO_2_ nanoheterostructure samples. Notably, the TiV-3 sample (atomic ratio of V/Ti approximately 1:1) showed strong absorbance in the visible-light region and had large interface heterojunctions that displayed the highest degradation rate of MB up to 24% compared to pure TiO_2_ films. The higher electronic conductivity which arise from greater portion of V^4+^/V_toatal_ supports the more active surface reaction and promoting the separation and transfer of photogenerated charge carriers. Thus, this research provides an alternative route for the one-step synthesis of porous V_2_O_5_/TiO_2_ nanoheterostructure films and shows that these nanocomposite films can potentially be used in photocatalytic applications.

## Figures and Tables

**Figure 1 molecules-25-03327-f001:**
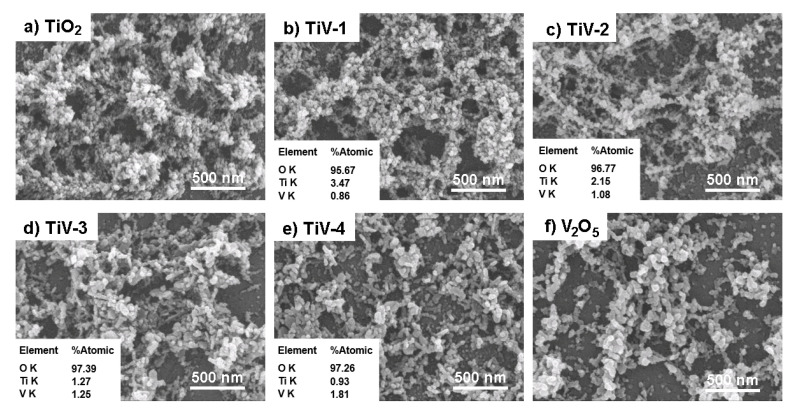
Surface morphologies of (**a**) TiO_2_, (**b**–**e**) TiV1–4, and (**f**) V_2_O_5_ samples.

**Figure 2 molecules-25-03327-f002:**
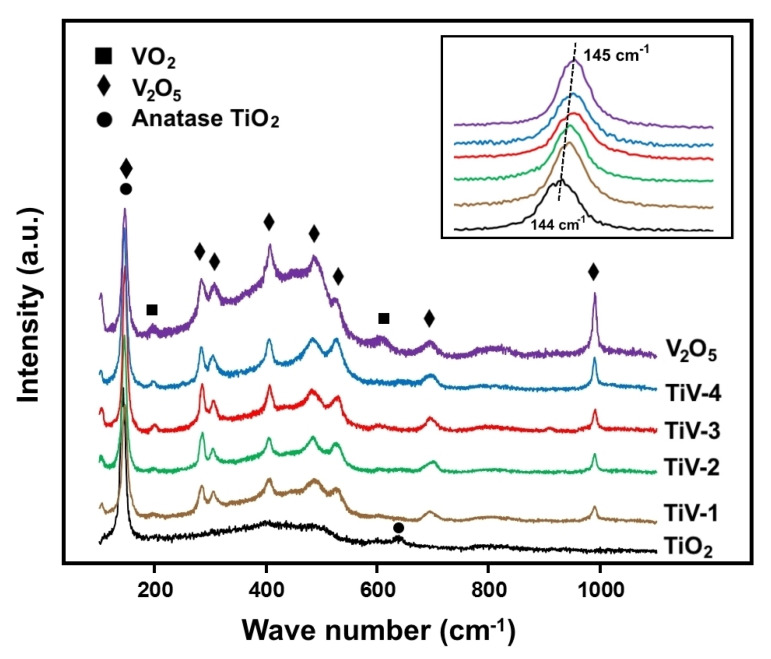
Raman spectra of pure TiO_2_, V_2_O_5_, and TiV1–4 composite samples.

**Figure 3 molecules-25-03327-f003:**
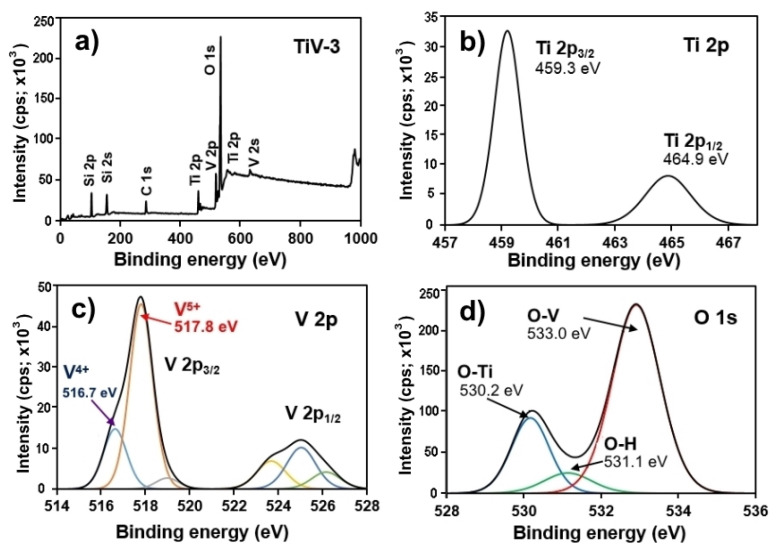
Spectrum of X-ray photoelectron spectroscopy for the (**a**) survey scan, and (**b**–**d**) high-resolution XPS spectra of the TiV-3 sample.

**Figure 4 molecules-25-03327-f004:**
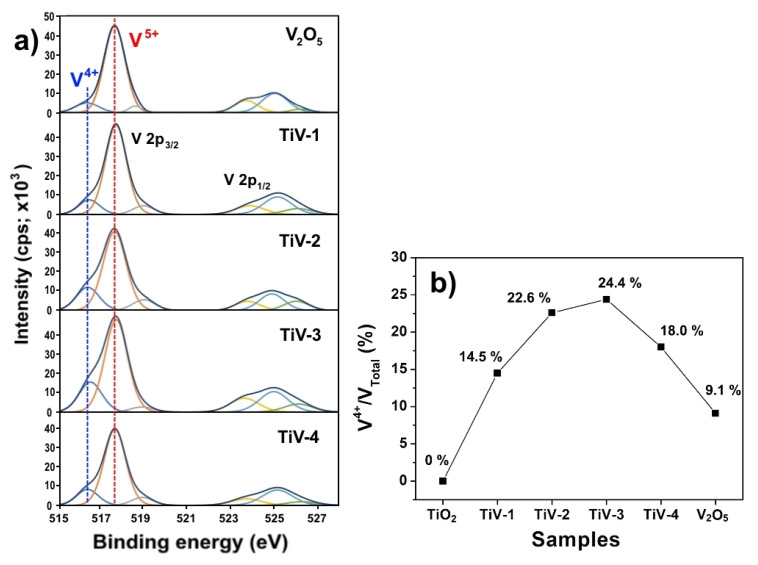
(**a**) A comparison of the high-resolution V 2p XPS spectra for all prepared samples and (**b**) V^4+/^V_total_ ratios of pure TiO_2_, TiV1-4 heterostructure, and V_2_O_5_ samples.

**Figure 5 molecules-25-03327-f005:**
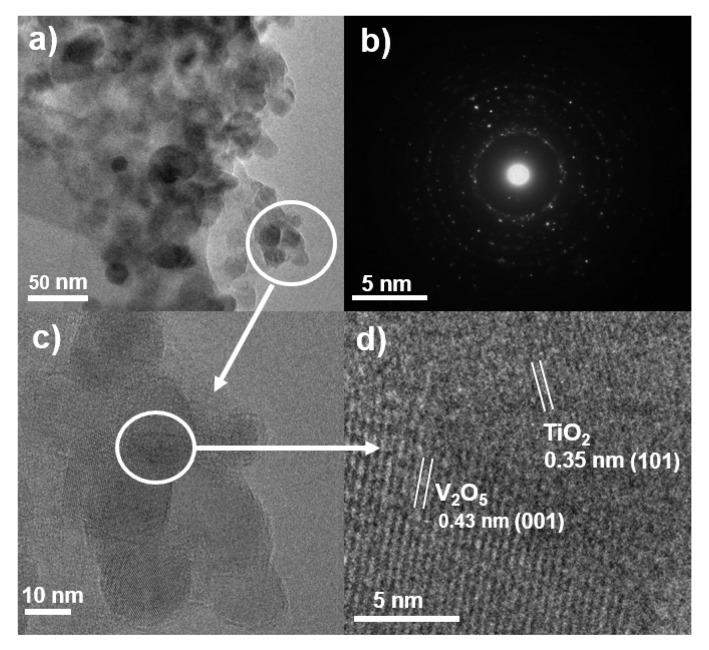
(**a**) Transmission electron microscope image, (**b**) selected area electron diffraction pattern, (**c**) high-resolution transmission electron microscopy image, and (**d**) partial enlargement of the TiV-3 sample.

**Figure 6 molecules-25-03327-f006:**
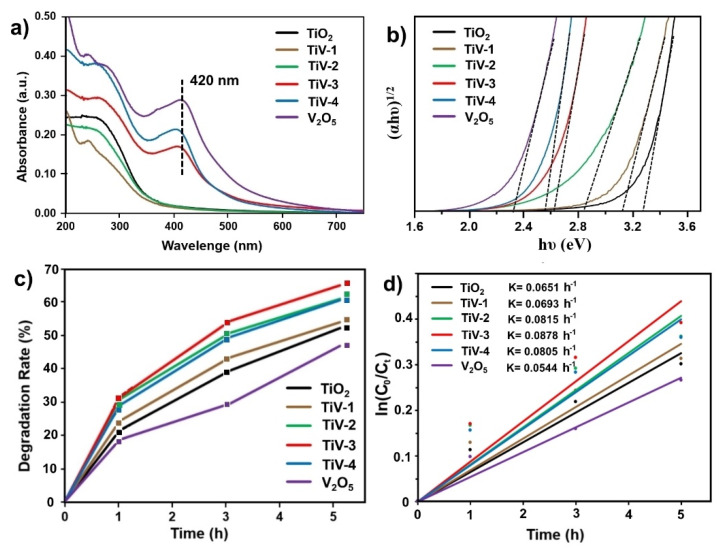
(**a**) UV-Vis absorption spectra, (**b**) Tauc plot, (**c**) Photocatalytic degradation, and (**d**) comparison of the rate constants of MB degradation over samples.

**Figure 7 molecules-25-03327-f007:**
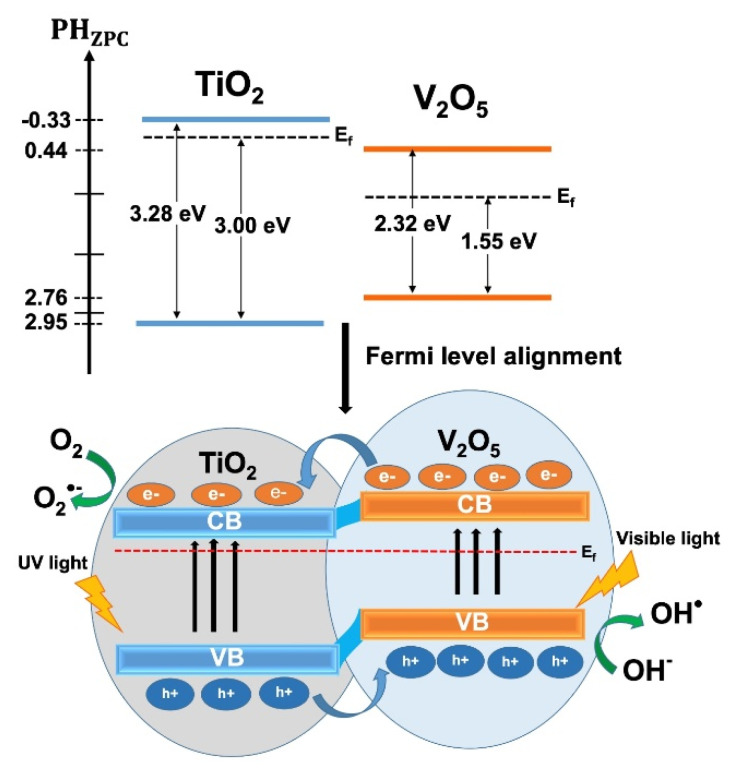
Possible mechanisms for photo-catalysis over the nanoheterostructure films.

**Figure 8 molecules-25-03327-f008:**
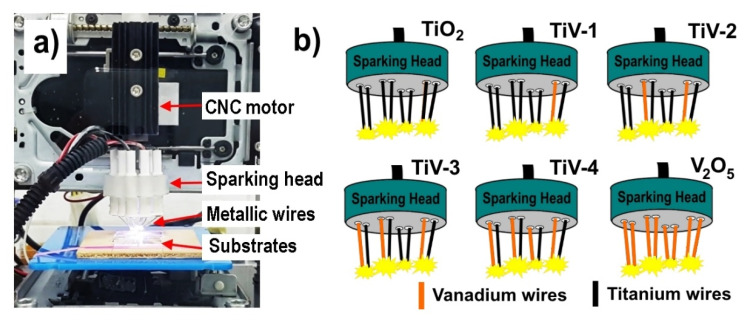
(**a**) Photo of the sparking apparatus and (**b**) schematic of sparking head of samples.

**Table 1 molecules-25-03327-t001:** Correlations of the atomic ratio of Ti/V, particle size, band gap, and the degradation rate of samples.

Samples	Atomic Ratio of Ti/V	Particle Size (nm)	Band Gap (eV)	V^4+^/V_total_ (%)	Degradation Rate (%)
TiO_2_	-	20 ± 2	3.28	0	52.0
TiV-1	4:1	35 ± 3	3.12	14.5	53.8
TiV-2	2:1	42 ± 5	2.84	22.6	60.8
TiV-3	1:1	48 ± 2	2.63	24.4	64.8
TiV-4	1:2	55 ± 3	2.56	18.0	60.4
V_2_O_5_	-	70 ± 4	2.32	9.1	46.6

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
