# Peer review of "Porous V_2_O_5_/TiO_2_ Nanoheterostructure Films with Enhanced Visible-Light Photocatalytic Performance Prepared by the Sparking Method"

_molecules, 2020, doi:10.3390/molecules25153327_

Round 1
Reviewer 1 Report
Porous V2O5/TiO2 nanoheterostructures films with different atomic ratios of Ti/V have been prepared to show enhanced photoactivity. Some interesting results are given. I would like to recommend it to publish after proper revisions.
- The recycling experiments should be done to show the photostability of composite photocatalysts.
- The formation of more O2•− radicals over V2O5/TiO2 lacks the support of sufficient experimental evidence.
- If using only visible light, how about the photocatalytic performance?
- When reporting the photocatalytic acitivty, it is highly suggested to show the details of light intensity.
- Related composite based photocatalysis is largely missed.
J.Mater.Chem.A, 2017,5, 23681.
Chem.Rev. 2015, 115, 10307-10377.
Author Response
Response to Reviewer 1 Comments
Porous V2O5/TiO2 nanoheterostructures films with different atomic ratios of Ti/V have been prepared to show enhanced photoactivity. Some interesting results are given. I would like to recommend it to publish after proper revisions.
Point 1: The recycling experiments should be done to show the photostability of composite photocatalysts.
Response 1: Thank you for your suggestion, the recycling performance should be investigated. However, due to time constraints, we are unable to do additional experiments.
Point 2: The formation of more O2•− radicals over V2O5/TiO2 lacks the support of sufficient experimental evidence.
Response 2: We believe that the existence of V4+ leads to higher photocatalytic performance by the higher formation of O2•− radicals. However, our experiment results didn’t have the evidence to support the increasing of O2•− radicals. We revised our manuscript (line 5-8 and 25-27 page 9 as highlighted) that the presence of V4+ as evidenced by the XPS result in our work which supports and provides the higher active sites. It also provides high electrical conductivity, leading to the boosting the separation and transfer of photogenerated charge carriers, which is the important for photocatalytic reactions, as reported previously [40-43].
Point 3: If using only visible light, how about the photocatalytic performance?
Response 3: The photocatalytic performance results in this paper (Figure 7c-d) were performed only in visible light.
Point 4: When reporting the photocatalytic activity, it is highly suggested to show the details of light intensity.
Response 4: According to the reviewer suggestion, we added the details about light intensity (~100 w/m2) in our revised manuscript.
Point 5: Related composite based photocatalysis is largely missed. J.Mater.Chem.A, 2017,5, 23681. Chem.Rev. 2015, 115, 10307-10377.
Response 5: Thank you for the reviewer suggestion, the two publications were added to the references of this manuscript.

Reviewer 2 Report
The article “Porous V2O5/TiO2 nanoheterostructures films with enhanced visible-light photocatalytic performance prepared by sparking method” investigate porous V2O5/TiO2 nanoheterostructures film for photocatalytic application. The synthesized V2O5/TiO2 by sparking method shows the higher photocatalytic activity than pristine TiO2 and the authors argue that V4+ ions in structure give an important role for enhanced activity. The development of TiO2 photocatlayst is still important issue and the reviewer consider it is interesting that they successfully synthesized V2O5/TiO2 by sparking method with improved activity. Thus, this manuscript can be accepted after some minor revisions as follows.
- In figure 4, y-axis’s unit is cps, but no value is shown. please correct its unit arbitrary unit or show cps values at y-axis.
- In figure 4, what components are other peaks (~100eV and ~150 eV) indicating?
- Figure 5 may be plotted by XPS results for all prepared sample. If then, the authors should provide actual XPS data. The reviewer wants to check deconvoluted peak for V4+ is well matched for all samples.
- In figure 8, the schematics indicate the oxidation and reduction occur at all sides (TiO2 and V2O5) at the same time. It does not make sense. If holes/electrons are separated as proposed by authors, reduction at TiO2 and oxidation at V2O5 should occur, respectively. Can the authors explain regarding this? or correct it.
Author Response
Response to Reviewer 2 Comments
The article “Porous V2O5/TiO2 nanoheterostructures films with enhanced visible-light photocatalytic performance prepared by sparking method” investigate porous V2O5/TiO2 nanoheterostructures film for photocatalytic application. The synthesized V2O5/TiO2 by sparking method shows the higher photocatalytic activity than pristine TiO2 and the authors argue that V4+ ions in structure give an important role for enhanced activity. The development of TiO2 photocatlayst is still important issue and the reviewer consider it is interesting that they successfully synthesized V2O5/TiO2 by sparking method with improved activity. Thus, this manuscript can be accepted after some minor revisions as follows.
Point 1: In figure 4, y-axis’s unit is cps, but no value is shown. Please correct its unit arbitrary unit or show cps values at y-axis.
Response 1: The figures 4(a-d) were revised, the unit of y-axis was added.
Point 2: In figure 4, what components are other peaks (~100eV and ~150 eV) indicating?
Response 2: Figure 4(a) was modified by added more details according to the reviewer suggestions. The peaks at ~100 eV and ~150 eV indicated components of the quartz substrate (Si 2p and Si 2s, respectively).
Point 3: Figure 5 may be plotted by XPS results for all prepared sample. If then, the authors should provide actual XPS data. The reviewer wants to check deconvoluted peak for V4+ is well matched for all samples.
Response 3: The deconvoluted peak for V4+ of all prepared samples was added to the revised manuscript (figure 5a).
Point 4: In figure 8, the schematics indicate the oxidation and reduction occur at all sides (TiO2 and V2O5) at the same time. It does not make sense. If holes/electrons are separated as proposed by authors, reduction at TiO2 and oxidation at V2O5 should occur, respectively. Can the authors explain regarding this? or correct it.
Response 4: Thank you for the reviewer suggestion. We agree with the reviewer, so the figure 8 was corrected.

Reviewer 3 Report
This paper deals with a valuable topic for technologic applications and is worth for publication for this reason. The structure of the paper is good, the methodologies are correctly described, and the results seem reliable. well-written and clear, in a correct English, and deals with. However, the English is very poor, and the paper absolutely needs to be corrected by an English native. Besides this general comment, I address below some details I noticed along the paper.
Introduction: an article about the photocatalytic properties of porous TiO2 thin films has been published this year. It should be worth to cite it: Photocatalytic properties of atomic layer deposited TiO2 inverse opals and planar films for the degradation of dyes. Birnal P. et al, Appl. Surf. Sci. 512 (2020)
Samples: the Ti/V atomic ratios for the 4 samples (TiV-1 to TiV-4) are not straightforward for me. In the abstract, it is written that the ratios equal 4:1, 2:1, 1:1, and 1:2. Lines 71-73, it is written that the vanadium concentration was varied by replacing titanium wire with vanadium wire from 1 pair to 4 pairs. Therefore, if one wants to recover the atomic ratios cited in the abstract by introducing 1, 2, 3 and 4 pairs of vanadium successively, the number of Ti wires needs to be 4, 4, 3, 2, which involves a change in the total number of wires from 5 for TiV-1 to 6 for the three other samples. And all this does not correspond to the scheme in Figure 1, where the total number of wires is 8 and the Ti/V ratios are 7/1 3/1 5/3 and 1/1. Please clarify.
Characterization methods: There is no detail about XPS (X-ray source? Type of analyser?). what was the laser power of the Raman spectrometer?
Photocatalytic activity measurement: I guess that the MB amount is given as a concentration i.e., 0.01 mmol/L and not as a quantity in mmol?
Figure 2: I am surprised by the very low Ti and V amounts compared to oxygen given by the EDS analyses. For instance, the TiV-3 sample should contain 10 at.% Ti, 20 at.% V and 70 at.% O. I reckon EDS is not perfectly quantitative, but the difference is huge (one order of magnitude).

Author Response
Response to Reviewer 3 Comments
Point 1: This paper deals with a valuable topic for technologic applications and is worth for publication for this reason. The structure of the paper is good, the methodologies are correctly described, and the results seem reliable. Well-written and clear, in a correct English, and deals with. However, the English is very poor, and the paper absolutely needs to be corrected by an English native. Besides this general comment, I address below some details I noticed along the paper.
Response 1: Thank you for your suggestion. Now, this manuscript (ID: molecules-862435, Title: Porous V2O5/TiO2 nanoheterostructures films with enhanced visible-light photocatalytic performance prepared by the sparking method) was completely English edited from the MDPI English Services.
Point 2: Introduction: an article about the photocatalytic properties of porous TiO2 thin films has been published this year. It should be worth to cite it: Photocatalytic properties of atomic layer deposited TiO2 inverse opals and planar films for the degradation of dyes. Birnal P. et al, Appl. Surf. Sci. 512 (2020)
Response 2: Thank you for your suggestion, this paper was added to our reference.
Point 3: Samples: the Ti/V atomic ratios for the 4 samples (TiV-1 to TiV-4) are not straightforward for me. In the abstract, it is written that the ratios equal 4:1, 2:1, 1:1, and 1:2. Lines 71-73, it is written that the vanadium concentration was varied by replacing titanium wire with vanadium wire from 1 pair to 4 pairs. Therefore, if one wants to recover the atomic ratios cited in the abstract by introducing 1, 2, 3 and 4 pairs of vanadium successively, the number of Ti wires needs to be 4, 4, 3, 2, which involves a change in the total number of wires from 5 for TiV-1 to 6 for the three other samples. And all this does not correspond to the scheme in Figure 1, where the total number of wires is 8 and the Ti/V ratios are 7/1 3/1 5/3 and 1/1. Please clarify.
Response 3: We don 't make sentence clear enough therefore the reviewer has misunderstood. TiV-1 to TiV-4 are only the name of the experimental conditions as shown in fig. 1. The number 1-4 are the number of vanadium wires replaced titanium wires from 1 pair to 4 pairs in the experiment apparatus whereas the ratios 4:1, 2:1, 1:1, and 1:2 in the abstract are the atomic ratio of samples which were observed by EDS as shown in fig 2. The sample names in figure 1 are not same of the atomic ratio.
Point 4: Characterization methods: There is no detail about XPS (X-ray source? Type of analyser?). What was the laser power of the Raman spectrometer?
Response 4: We added more detail about the X-ray source (Al K-alpha, 1487 eV) and laser power of Raman spectrometer (150 mW) to our revised manuscript.
Point 5: Photocatalytic activity measurement: I guess that the MB amount is given as a concentration i.e., 0.01 mmol/L and not as a quantity in mmol?
Response 5: Thank you for the reviewer comment. We edited the concentration unit of MB to mmol/L.
Point 6: Figure 2: I am surprised by the very low Ti and V amounts compared to oxygen given by the EDS analyses. For instance, the TiV-3 sample should contain 10 at.% Ti, 20 at.% V and 70 at.% O. I reckon EDS is not perfectly quantitative, but the difference is huge (one order of magnitude).
Response 6: The sparking films in this experiment were very thin and porous therefore the large amounts of oxygen peak in EDS results is due to the quartz substrates (SiO2) as show in figure 4(a) (page12 line 40-42 as highlighted).

Round 2
Reviewer 1 Report
publish as it is